# Deep learning-based automatic image classification of oral cancer cells acquiring chemoresistance in vitro

Hsing-Chuan Hsieh[1], Cho-Yi Chen[2], Chung-Hsien Chou[3], Bou-Yue Peng[4,5], Yi-Chen Sun[6,7], Tzu-Wei Lin[8], Yueh Chien[8], Shih-Hwa Chiou[8], Kai-Feng Hung[8,9]*, Henry Horng-Shing Lu[1,8,10,11]*

1 Institute of Statistics, National Yang Ming Chiao Tung University, Hsinchu, Taiwan, 2 Institute of Biomedical Informatics, National Yang Ming Chiao Tung University, Taipei, Taiwan, 3 Institute of Oral Biology, National Yang Ming Chiao Tung University, Taipei, Taiwan, 4 Department of Dentistry, Taipei Medical University Hospital, Taipei, Taiwan, 5 School of Dentistry, College of Oral Medicine, Taipei Medical University, Taipei, Taiwan, 6 College of Medicine, Tzu-Chi University, Hualien, Taiwan, 7 Department of Ophthalmology, Taipei Tzu Chi Hospital, The Buddhist Tzu Chi Medical Foundation, New Taipei, Taiwan, 8 Department of Medical Research, Taipei Veterans General Hospital, Taipei, Taiwan, 9 Department of Dentistry, School of Dentistry, National Yang Ming Chiao Tung University, Taipei, Taiwan, 10 School of Post-Baccalaureate Medicine, Kaohsiung Medical University, Kaohsiung, Taiwan, 11 Department of Statistics and Data Science, Cornell University, Ithaca, New York, United States of America

* kfhung@vghtpe.gov.tw (KFH); henryhslu@nycu.edu.tw (HHSL)

**Data Availability Statement:** All relevant data for this study are publicly available from the Kaggle repository (https://www.kaggle.com/datasets/

## Abstract

Cell shape reflects the spatial configuration resulting from the equilibrium of cellular and environmental signals and is considered a highly relevant indicator of its function and biological properties. For cancer cells, various physiological and environmental challenges, including chemotherapy, cause a cell state transition, which is accompanied by a continuous morphological alteration that is often extremely difficult to recognize even by direct microscopic inspection. To determine whether deep learning-based image analysis enables the detection of cell shape reflecting a crucial cell state alteration, we used the oral cancer cell line resistant to chemotherapy but having cell morphology nearly indiscernible from its non-resistant parental cells. We then implemented the automatic approach via deep learning methods based on EfficienNet-B3 models, along with over- and down-sampling techniques to determine whether image analysis of the Convolutional Neural Network (CNN) can accomplish three-class classification of non-cancer cells vs. cancer cells with and without chemoresistance. We also examine the capability of CNN-based image analysis to approximate the composition of chemoresistant cancer cells within a population. We show that the classification model achieves at least 98.33% accuracy by the CNN model trained with over- and down-sampling techniques. For heterogeneous populations, the best model can approximate the true proportions of non-chemoresistant and chemoresistant cancer cells with Root Mean Square Error (RMSE) reduced to 0.16 by Ensemble Learning (EL). In conclusion, our study demonstrates the potential of CNN models to identify altered cell shapes that are visually challenging to recognize, thus supporting future applications with this automatic approach to image analysis.

janehsieh/oral-cancer-cells-with-chemoresistance-in-vitro/data).

**Funding:** This study was financially supported by the National Science and Technology Council in the form of grants (112-2634-F-A49-003-, 113-2118-M-A49-007-MY2, 113-2923-M-A49-004-MY3) received by HHSL. This study was also financially supported by the Taipei Veterans General Hospital in the form of grants (V113C-050, V113D72-002-MY2-1) received by KFH. This study was also financially supported by the Higher Education Sprout Project of the National Yang Ming Chiao Tung University in the form of an award from the Ministry of Education, Taiwan received by HHSL.

**Competing interests:** The authors have declared that no competing interests exist.

## Introduction

Cancer growth is often challenged by different physiological and microenvironmental impacts, including hypoxia and nutrient deprivation, which are overcome by cell adaptation, a dynamic process of cell state transition, to enhance its survival under various cellular stress [1–4]. Among various parameters depicting the transition of the cell state, cell shape reflects the spatial configuration resulting from the equilibrium of cellular and environmental signals [5, 6]. Consequently, cell shape has been considered a highly relevant indicator of its function and biological properties [7, 8]. Indeed, as exemplified by the epithelial-to-mesenchymal transition (EMT) process, cancer cells that have lost typical epithelial characteristics become more invasive and less responsive to chemotherapy [9]. In addition, it has been well recognized that undifferentiated cancer cells manifest different morphological characteristics and behave more aggressively [10, 11], and that highly metastatic cancer cells exhibit more prominent morphological changes than their non-metastatic counterparts [12]. Moreover, several studies showed that acquisition of chemoresistance was accompanied by significant morphological changes [12, 13], albeit indiscernible shape alteration was also reported in cells adaptive to low-dose chemotherapy [14]. As such, it is plausible to reveal cell state transitions using cell shape monitoring.

Convolutional neural network (CNN) is a class of Artificial Neural Networks (ANN) known for processing (but not limited to) visual imagery data such as images and videos [15]. Their great success in the application includes image classification, object detection (like faces in photos), and image segmentation [16–18]. In general, ANN mimics how the networks of biological neurons in the brain work [19]. The classical ANN is composed of many layers from one input layer (receptor), one or more hidden layers (subsequent processers), and one final output layer (reactor). Each layer consists of different numbers of artificial neurons (i.e., nodes), which connect to all nodes of the previous layer by a series of mathematical weights that resemble the synapses between neurons. ANN learns from data fed into the input layer. The hidden layer then multiplies the input values by the initial weights of the connections for each node and sums up. Once the sums reach the respective thresholds, the corresponding nodes are activated and transmit the processed signal value to the nodes of the next layer. The activation function in ANN controls the activation threshold of each node. Furthermore, the way the next hidden layer receives and processes the incoming signals is essentially the same, except for a different activation design or the number of nodes. This process repeats until the output layer, of which the estimated answer comes out. The error is then computed between the estimate and the true answer (that is, an exact match for the input), and the network model would adjust the weights of connections for every layer until the prediction errors are minimized. This try-and-error learning process in ANN is called model training or fitting.

While CNN has been applied to identify patterns and structures in images [16–18, 20–22], a challenge is that the transition from one state to another is a continuum, and so is the alteration of cell shape. As has been well characterized in studies of EMT and its reverse process (MET), the transition between the epithelial and mesenchymal states is not binary. In fact, a hybrid or intermediate epithelial / mesenchymal phenotype is common [23–25]. Meanwhile, cancers often evolve with spatial and temporal heterogeneity in cell size, morphology, growth, and physiology, thus exhibiting a broad spectrum of intermediate phenotypes within the population [26, 27]. Indeed, many cancers, including oral cancer, typical features include polygonal, ovoid, or elongated morphology, characterized by variation in cell size, hyperchromatic nuclei resulting from increased DNA content due to abnormal division, and higher nuclear-to-cytoplasmic ratio with enlarged nuclei relative to the cytoplasm. As cancer cells progress to more malignant states, the nuclei may become larger and irregular and the morphology often adopts

a spindle shape, indicative of EMT. These characteristics complicate the assessment of the cell state by image analysis. Since CNN excels in automatically learning and extracting features from images, this model could represent an exceptional tool for detecting cell state transition. Moreover, image analysis enables the determination of cell states without cell processing, thus extending its applicability to live cells [28], Therefore, it is of importance to determine and leverage the capability of CNN in in handling complex medical imaging tasks.

Taken together, in this study, we aim to explore the ability of CNN to detect a subtle change in cell shape across different cell states, which has not been clearly demonstrated. We provide evidence to support the fact that CNN represents an ideal tool for predicting the cell transition state even if its morphological change is not visually appreciable.

## Materials and methods

### Data sets preparation

**Data description.** To determine whether CNN-based analysis can identify cell morphological changes after acquiring chemoresistance, we treated OECM1 oral cancer cells (hereafter named "PARENTAL") with thapsigargin at a tolerable level to induce a cellular stress response that has been shown to enhance their resistance to cisplatin. Importantly, these chemoresistant OECM1 cells (named "RESISTANT") exhibit subtle, if not absent, alteration in cell shape, which is not visually appreciated under a microscope. Besides, we also used normal oral keratinocytes (named "CONTROL") as a non-cancer control in this study.

**Cell culture and image acquisition.** The OECM1 cell line derived from a patient with gingiva squamous cell carcinoma was grown in Roswell Park Memorial Institute Medium (RPMI; 11-100, Biological Industries) supplemented with 10% Fetal Bovine Serum (FBS; 10438-028, ThermoFisher Scientific), 1% Penicillin-Streptomycin (15140-122, ThermoFisher Scientific), and 1% glutamine [29]. To generate the chemoresistant line through adaptation to cellular stress at a tolerable level, $5 \times 10^4$ OECM1 oral cancer cells were cultured for 4 days in media containing 2.5 μM thapsigargin, a known stressor for the endoplasmic reticulum (ER). This approach has previously been shown to increase the IC50 of cisplatin chemotherapy for OECM1 cells from 25 μM to 50 μM, as determined by a colorimetric NAD(P)H-based assay to measure the activity of dehydrogenase, a mitochondrial enzyme active only in live cells, suggesting the development of chemoresistance [30]. As a result, a total of 900 images of untreated OECM1 cells ("PARENTAL"), chemoresistant OECM1 cells ("RESISTANT"), and normal oral keratinocytes ("CONTROL") were taken with a benchtop microscope (Fig 1). Images of a single population of parental OECM1 cells, chemoresistant OECM1 cells, and normal oral keratinocytes were classified as homogeneous data sets (HOMO).

To simulate heterogeneity within tumors, PARENTAL and RESISTANT cells were harvested, mixed in a 1:2, 2:1, 1:3, or 3:1 ratio, plated in a 6-cm dish, and incubated at 37˚C for 16 hours before image acquisition. Five dishes of cells from each condition were prepared as biological replicates. Forty to sixty images of each dish were randomly acquired using a microscope (Leica DMi8) with a bright field objective of magnification of 100X. As a result, 200 images of each ratio were collected, resulting in a total of 800 images. The images of mixed cells (i.e., featuring soft labels) were classified as the heterogeneous data set (HETERO).

**Preprocessing.** All cell images (including HETERO data) were saved in TIFF format (extension:.tif), with a size (that is, (height, width, channel)) of (2048, 2880, 3). To comply with the requirement of the model input (i.e., EfficientNet-B3), these images were resized to (300, 300, 3), as demonstrated in Fig 1.

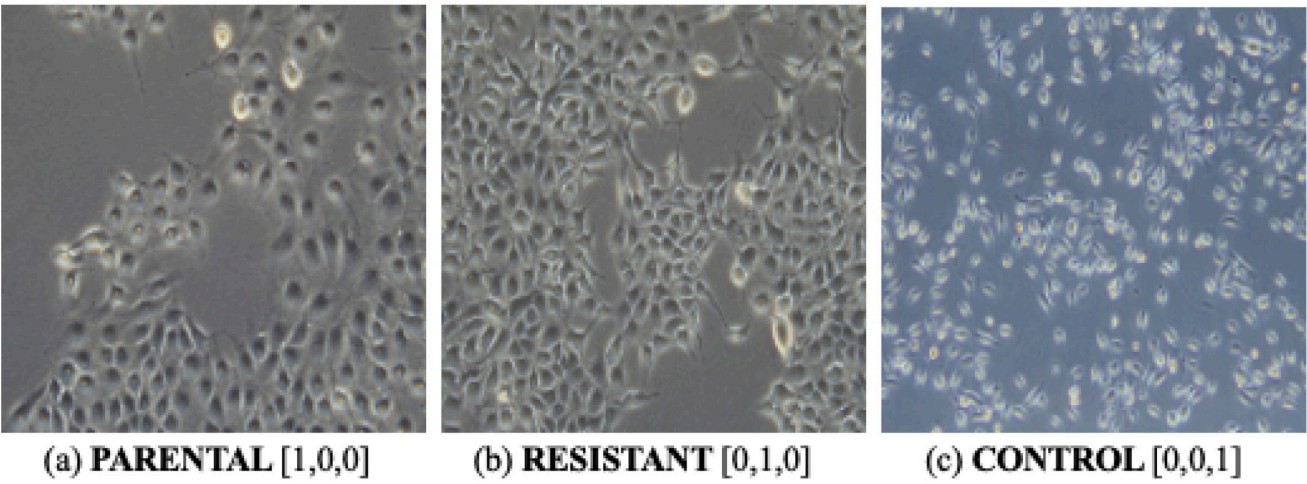

**Fig 1. Three classes of cells (with labeling), named as HOMO data.** Each class has equally 300 images. Notice that the morphological features of three classes of cells have only subtle differences that the expert might be able to distinguish.

## Experimental procedure

After basic preprocessing, our study then used the HOMO data set to examine whether CNN-based image analysis can perform the three-class classification of cells with or without chemoresistant morphology. We also used the HETERO data set to determine the proficiency of such analysis in approximating the ratio of PARENTAL and RESISTANT cells in a population. Fig 2 briefly shows our experimental framework for model learning and evaluation; further details of the procedures were explained in the following sections.

**K-fold cross-validation.** First, we spared 20% of the HOMO data as a test set that would not be used for model training; we performed K-fold cross-validation (CV) on the rest of the HOMO data. K-fold CV is a resampling procedure in which the data are randomly divided into K groups (K = 5 in our case); each group is used in turn as a validation set, while the rest of the K-1 groups are used as a training set for network learning. Consequently, a total of K models would be trained and their average performance in all validation sets could serve as an index for model evaluation as well as hyperparameter tuning. Finally, training models with the best hyperparameters would apply to the set of tests for prediction evaluation.

In addition, to show the capacity of the HETERO data, we would like to compare the model performances with or without HETERO data. To do so, we also divide the HETERO data into training, validation, and test sets with a percentage of 64:16:20. Due to the limited samples of HETERO data, their training and validation sets would be only randomly split once, instead of five-fold CV splits as in HOMO data. As a result, in addition to the model trained with the HOMO training data (from the five-fold CV), a corresponding model was also trained with the same HOMO training set plus the HETERO training set. To compute the average performance for the models trained with additional HETERO data, each training model from the five-fold CV was again evaluated with the corresponding HOMO validation set, as well as with the HETERO validation set. Note that the evaluation metrics for the HOMO data (with hard labels) were different from those for the HETERO data (with soft labels), which will be introduced later. Finally, the trained-with-HETERO model with the best hyperparameters (that is, the one with the optimum metric value for HETERO data) would again apply to the test set of HOMO data as well as HETERO data, for prediction evaluation.

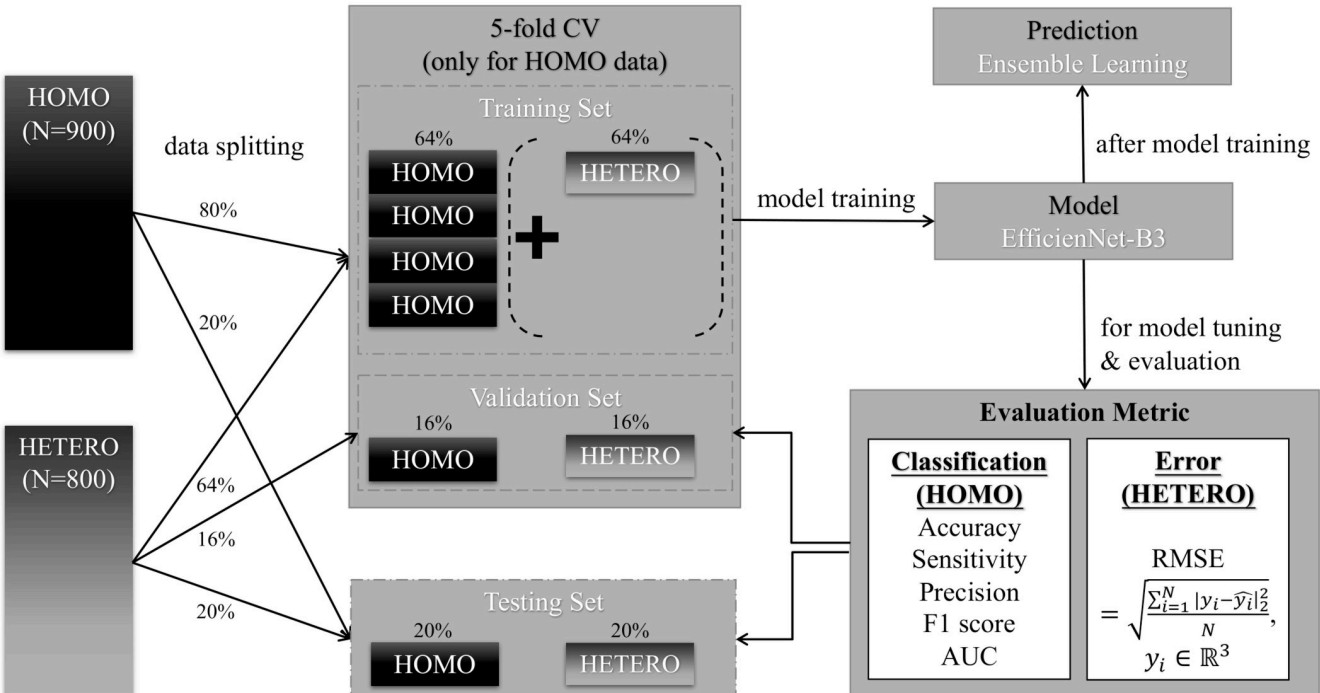

**Fig 2. Experimental framework.** The model were trained with both HOMO and HETERO data. For HOMO data, 20% of random data are spared as test set, and the rest of data were applied with five-fold CV. For HETERO data, the data were directly split into test (20%), training (64%) and validation sets (16%) without CV. To evaluate the model, classification metrics were applied to HOMO data, while the error metric for approximation to HETERO data. For prediction, ensemble learning is applied to aggregate all training model results from five-fold CV.

**Training set preprocessing.** Due to the imbalanced issue and limited sample size for each type of HETERO data compared to HOMO data, we further preprocessed the HETERO training sets prior to model training, in order to facilitate the model training with the HETERO data set. Therefore, we proposed OS techniques to generate more HETERO data. These include, but are not limited to, the following methods.

- Random OS: Random OS is the simplest oversampling method, in which we randomly resample data from the minority class until its sample size is as balanced as that of the majority class. Here we up-sample each type of HETERO data and tune their sample sizes via random search, in order to optimize the model performance on HETERO data.

- SMOTE: As there are no same fingerprints across humans, there are also no same instances across data. SMOTE provides another popular way to synthesize new data [31], rather than simply duplicating randomly selected data as OS does. To synthesize artificial but somewhat realistic data, SMOTE at first randomly chooses a sample from the minority class (regarded as a reference), then its K nearest neighbors (either restricting to the same or different class) can be identified using the K Nearest Neighbors (KNN) method (default: K = 3). Finally, random linear interpolation is executed on the reference data point and one of its neighbors, to generate new data. In our application, the reference data are different types of the HETERO data—1:2, 2:1, 1:3, and 3:1; their corresponding neighboring data are then taken from their adjacent classes, which are 1:3, 3:1, RESISTANT and PARENTAL (the latter two types are from the HOMO data), respectively. A demonstration of our SMOTE operation can be seen in Fig 3.

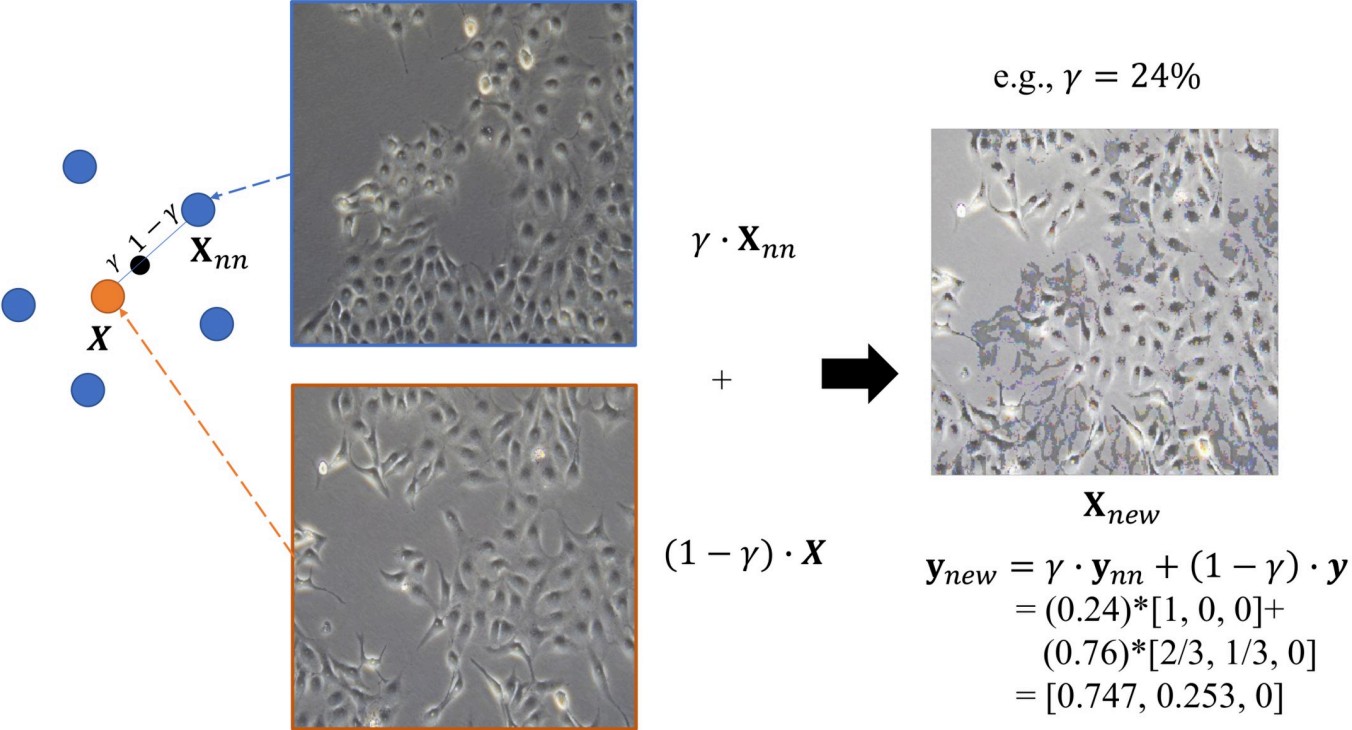

**Fig 3. Illustration of SMOTE to generate soft-label data between [1, 0, 0] $\sim$ [2/3, 1/3, 0].** In this example, the reference data point is [2/3, 1/3, 0] (orange) with its $k$ = 5 nearest neighbors of counter class—[1, 0, 0] (blue). To generate synthetic data ($\mathbf{X}_{new}$, $\mathbf{y}_{new}$), a linear interpolation is operated between reference point and one of its nearest neighbors via random parameter $\gamma \in$ [0, 1].

- Borderline-SMOTE: Either the random OS or the SMOTE methods can guarantee whether the resampled or reference data are crucial or trivial. Usually, the data at the borderline of different classes are more crucial than the rest of the data, since a clearer borderline can help the classifier draw a more robust decision boundary and hence improve the classification power. Therefore, we also applied Borderline-SMOTE [32], where only minority examples near the borderline have been oversampled and synthesized (same as SMOTE, but with their neighbors restricted to the same minority class), in order to improve the quality of the reproduced data (Fig 4). Furthermore, to generate more data for each type of HETERO data, the definition of the boundary is feasible only with the existence of another competing class of the data; just as introduced in the SMOTE method, we took adjacent types of HETERO data as the competing class. For the reference 1:2, 2:1, 1:3, and 3:1 HETERO data, we chose 1:3, PARENTAL, RESISTANT and PARENTAL, respectively, as their corresponding competing classes.

In addition, it has been shown that model performance can be improved by combining down-sampling (DS) and OS techniques [33], rather than using either method alone. In contrast to OS method, the idea of DS technique is to delete some data out of the majority class, in order to reach more balanced data (e.g., random DS—the opposite of random OS—randomly deletes the majority examples until the whole data are balanced). However, the question is which data should be deleted. In the cases of our HETERO data, due to the different growth speeds of cancer cells (PARENTAL vs. RESISTANT), it was expected that the proportions of

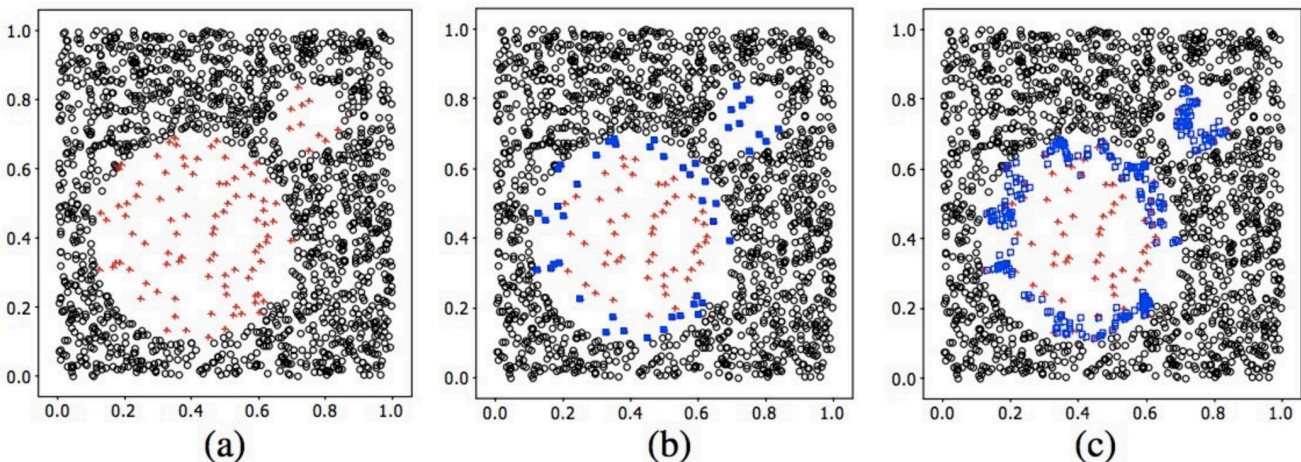

**Fig 4. Demonstration of Borderline-SMOTE on 2D data.** (a) The original distribution of Circle data set. (b) The borderline minority examples (solid blue squares). (c) The borderline synthetic minority examples (hollow blue squares). Source: [32].

mixed cells in a few images might actually be disproportionate, which unfortunately could not be observed by human eyes (e.g., a claimed "1:2" HETERO image could have an actual ratio closer to "1:3"). Such misleading data usually lie on the borders of different types of HETERO data and should be eliminated to decrease uncertainty. Accordingly, we propose the following DS technique:

- ENN: Since random DS methods cannot guarantee whether deleted data are crucial or not, and inadvertent deletion of crucial data causes more harm than trivial ones, the ENN method aims to remove ambiguous and noisy majority examples (e.g., those near the borderline or those beleaguered with minorities) using the KNN method [34]. More specifically, the ENN DS rule removes all (majority) instances which have been misclassified by the KNN method (default: K = 3) from the training set (as illustrated in Fig 5).

For the models that combined both OS and ENN DS methods, we performed the ENN method to the HETERO data in order to delete some ambiguous data first, before performing Random OS again to those kept HETERO data. The reason why the ENN technique was performed first was that by first eradicating the 'ambiguous' data points (usually lying on the borderline of different classes), it prevented the Random OS technique used later from duplicating potential questionable data, which would become another confounding factor. On the contrary, if Random OS was performed first and the duplicated data happen to be ambiguous, then ENN would only eradicate them again, which might undo the efficacy of OS.

**Model training.** As mentioned earlier, our goal was the three-class classification for hard-label HOMO data, as well as the probability prediction for soft-label HETERO data. For efficient transfer learning [35], we used EfficientNet published by Google [36] for model training. As the name suggested, it used relatively fewer parameters to achieve much better accuracy and efficiency than previous CNN models in ILSVRC [37]. EfficientNet includes a series of versions (B0 ∼ B7), each of which required different image resolution and hardware constraints; they thus benefited practitioners limited to different resources. We utilized version B3 as the discriminating pre-trained model, which was already sufficient for our tasks. The architecture of this model was kept, except for replacing the fully connected layer directly with a

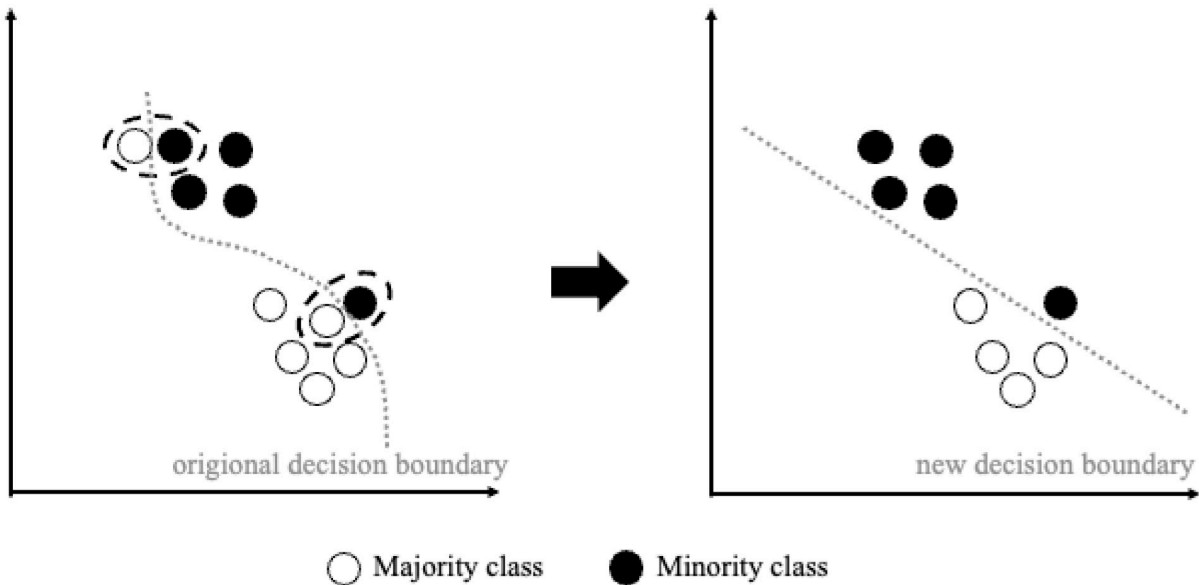

**Fig 5. Demonstration of ENN editing with 1-NN classifier on 2D data.** The left plot is the original data set, and the right one is the down-sampled data after using ENN (i.e., Edited data set); It shows that the data of majority class which have counter-class $k = 1$ nearest neighbor are deleted, which also make the decision boundary more robust (i.e., more linear). Source: plot adapted from [34].

ternary output layer, which predicted three-class probabilities via the SoftMax activation function. All of the model weights were discarded and retrained by the training images.

Besides, since HETERO data were also incorporated into model training, we rather considered the data labeling as a spectrum of probability distributions for different types of data. From this point of view, the labeling of the HETERO data was regarded as a continuous probability for each class (either PARENTAL or RESISTANT) of cells, and that of HOMO data only represented the extreme points of the continuum indicating that only one type of cells was distributed in the image. As a result, the loss function used for model optimization and weights estimation was Mean Squared Error (MSE), which was mainly developed for continuous data, instead of traditional Cross-Entropy (for categorical data). In other words, we took our task as a "regression task" rather than only a classification task.

**Evaluation metrics.**   To evaluate model training with different preprocessing techniques, five-fold CV was implemented, and the average performance and standard error (SE) of models were examined by the validation sets of the HOMO and HETERO data. There were two types of evaluation metrics for our tasks: categorical metrics for HOMO data with hard labels, and regression metrics for the HETERO data with soft labels. For hard-label data, we evaluated the model performance with macro version of Sensitivity, Precision, F1 score and AUC. These metrics originated from the evaluation of binary classification, and macro evaluation only exploits this concept by regarding each class as positive class while the rest of classes as negative in order to calculate the binary metrics for each class; a macro evaluation then takes the simple average of these metric values of all classes. Higher values (in proportion) of these metrics meant a better model performance. In addition, since we'd like to predict the true probability distribution of the HETERO data as close as possible, we therefore chose the well-known error metric—RMSE, defined as follows:

$$RMSE = \sqrt{\frac{\sum_{i=1}^{N} |\mathbf{y}_i - \hat{\mathbf{y}}_i|_2^2}{N}} \tag{1}$$

**Table 1. Hyperparameter tuning for training set preprocessing.**

| Preprocessing Methods | Range of $\Delta N$ | Interval | Hyperparameter values | | | |
|:---:|:---:|:---:|:---:|:---:|:---:|:---:|
| **OS** | [64, 384] | 64 | $\Delta N^{OS}_{120}$ | $\Delta N^{OS}_{210}$ | $\Delta N^{OS}_{130}$ | $\Delta N^{OS}_{310}$ |
| | | | 256 | 128 | 256 | 256 |
| **DS** | [-128, 0] | - | $\Delta N^{DS}_{120}$ | $\Delta N^{DS}_{210}$ | $\Delta N^{DS}_{130}$ | $\Delta N^{DS}_{310}$ |
| | | | -17 | 0 | -29 | -2 |

(1) OS techniques include Random OS, SMOTE and Borderline-SMOTE. (2) The hyperparameter values for DS were automatically determined by the ENN algorithm, but still presented here for comparison.

where $\mathbf{y}_i$, $\hat{\mathbf{y}}_i \in \mathbb{R}^3$ are the true and predicted labels (probabilities) of observation $i$. A higher value of RMSE meant higher error and a worse model performance.

**Hyperparameters summarization.** Since the model performance with HETERO data is as well in consideration, the metric mainly chosen for hyperparameter tuning is RMSE (Eq 1). In other words, the best hyperparameters are the ones that caused the training model to minimize the mean RMSE values calculated from five-fold CV.

As shown in Table 1, the number of increased data for each type of HETERO data (symbolized as $\Delta N^{OS}_{label}$, where $label$ = 120, 210, 130, 310) by OS techniques were the hyperparameters that could be tuned via five-fold CV. On the other hand, the numbers of eliminated data from ENN DS technique were automatically determined by the algorithm according to data distributions themselves and had no need to be tuned (but for convenience, we still symbolize them as $\Delta N^{DS}_{label}$ to show the results after ENN operation). We observed that the sample size of the training set for each type of the HETERO data is 128 (i.e., $200 \times 64\%$), which is slightly smaller than that of each class of the HOMO data (i.e., $192 = 300 \times 64\%$); the search domain for all hyperparameters $\Delta N^{OS}_{label}$ is then decided between the range of [64, 384] with an interval of 64. After the random search with five-fold CV, we found out that, for training models that incorporate Random OS, SMOTE or Borderline-SMOTE methods, the optimum over-sampled sample size for each type of HETERO data is $\{\Delta N^{OS}_{120} : 256, \Delta N^{OS}_{210} : 128, \Delta N^{OS}_{130} : 256, \Delta N^{OS}_{310} : 256\}$. For models that also incorporate DS method, in order to better eliminate the ambiguity between "1:2" and "2:1" HETERO data, ENN technique was applied and then 17 data points (i.e., $\Delta N^{DS}_{120}$ = -17) out of 128 instances of "1:2" training set were deleted (i.e., only 111 instances were kept for subsequent OS operation). Similarly, in order to better clarify the differences between "1:3" and "1:2" as well as between "3:1" and "2:1," ENN technique was also utilized to them, respectively. The reduced sample size turned out to be $\{\Delta N^{DS}_{130} : -29, \Delta N^{DS}_{310} : -2\}$ (i.e., 99 instances were kept for "1:3" data and 126 for "3:1," for subsequent OS operation). Furthermore, for subsequent OS reprocessing after ENN method, the remaining HETERO would be over-sampled again, of which the increased sample size of each type was based on the tuned value of $\Delta N^{OS}_{label}$ mentioned above. As a result, all models with reprocessing techniques (i.e., model M3 $\sim$ M5 introduced in "Results" section) could be compared to each other.

For hyperparameters included in CNN modling and image data augmentation (to expand the training data variation during model training), we empirically set the values as shown in Table 2. Note that while we set the number of epochs as 10, the learning rate for the first 4 epochs would be $7 \times 10^{-5}$, and $3 \times 10^{-5}$ afterward for the rest.

**Prediction with bagging ensemble learning.** Finally, in order to more accurately predict the class of new HOMO data and approximate the probability distribution for new HETERO data, we aggregated the predicted results from five training models (from five-fold CV) with

**Table 2. Hyperparameters setting for modeling.**

| Data augmentation | | Model | |
|---|---|---|---|
| Hyperparameter | Value | Hyperparameter | Value |
| rotation | 90˚ | learning rate | $7 \times 10^{-5}$ when epoch < 5, $3 \times 10^{-5}$ when epoch < 12 |
| horizontal flips | True | batch size | 16 |
| vertical flips | True | epochs | 10 |

the selected hyperparameters aforementioned, and eventually output a final result. Such method is called "bagging ensemble learning (EL)" [38]. More specifically, as Fig 6 suggested, after the input of new data (either HOMO or HETERO data), each model would predict the probabilities of 3 classes– [PARENTAL, RESISTANT, CONTROL] (i.e., $\hat{\mathbf{P}}_i^{(j)}, j = 1, \ldots 5$); then the final output of combinational probabilities out of 5 predictions could be derived either by simple or weighted average. For weighted average, the weights for each model could rely on the estimated accuracies or F1 scores computed from each fold of validation set. However, as shown in Table 3 (in the next section), the estimated accuracies and F1 scores were only minor differences between distinct folds of training models due to small SE values, here it sufficed to use simple average to generate the final prediction. Moreover, if the input image is HOMO, then the class with the largest predicted probability would be regarded as the predicted class for such image, this procedure is equivalent to "soft voting"; if the input image is HETERO, then the final aggregated distribution would be considered as its probable approximation (i.e., $\hat{\mathbf{y}}_i = \hat{\mathbf{P}}_i$). Consequently, the performance of our CNN-based image analysis on the three-class classification, as well as the approximation of the mixed population, were assessed using the HOMO and HETERO data set, respectively.

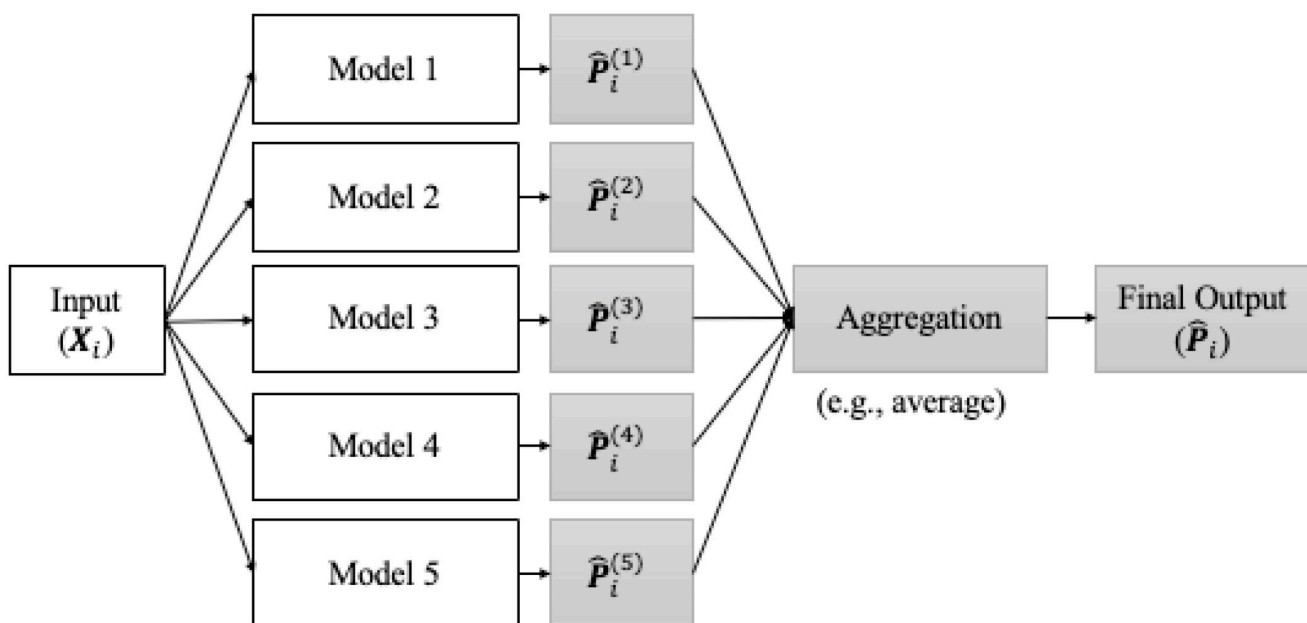

**Fig 6. Prediction via bagging ensemble learning.** The final prediction are based on the aggregation (here we use average) of the prediction results from five trained models (through five-fold CV).

**Table 3. The five-fold performances (Mean(SE)) of CNN models on the validation set of HOMO data.**

| Metrics (%) | Models | | | | |
|---|---|---|---|---|---|
| | **M1** | **M2** | **M3** | **M4** | **M5** |
| *Accuracy* | **99.58 (0.38)** | 99.17 (1.14) | 98.47 (1.58) | 98.33 (0.79) | 98.33 (1.81) |
| *Macro Sensitivity* | **99.58 (0.38)** | 99.17 (1.14) | 98.47 (1.58) | 98.33 (0.79) | 98.33 (1.81) |
| *Macro Precision* | **99.59 (0.37)** | 98.18 (1.12) | 98.50 (1.53) | 98.40 (0.71) | 98.45 (1.65) |
| *Macro F1 score* | **99.58 (0.38)** | 99.17 (1.14) | 98.47 (1.58) | 98.34 (0.78) | 98.33 (1.82) |
| *Macro AUC* | 99.95 (0.12) | **99.98 (0.03)** | **99.98 (0.03)** | 99.95 (0.03) | 99.88 (0.20) |

Values in **boldface** are the best result of the row.

## Results

To determine which data set serves best for model training, we prepared the following models with which the training sets were processed differently:

1. M1. Only the HOMO data set was used for model training.

2. M2. Both the HOMO and the HETERO data sets were used for model training.

3. M3. Both the HOME and the HETERO data sets were used. For the HETERO data set, we additionally performed Borderline-SMOTE to generate more synthetic HETERO data for model training. (Since the model results with the SMOTE technique is similar to that of M3, they are not shown here.)

4. M4. Both the HOME and the HETERO data sets were used. For the HETERO data set, we additionally performed Random OS to generate more HETERO data for model training.

5. M5. Both the HOME and the HETERO data sets were used. Additionally, we performed the ENN-DS method to delete some ambiguous cell images in HETERO data set first and then used the Random OS to generate more HETERO data. The basis underlying this data processing was that the elimination of ambiguous data may prevent the duplication of potential questionable cell images. On the other hand, when Random OS was performed first and ambiguous data was accidently duplicated, subsequent ENN would eliminate these images, thereby undermining the efficacy of OS.

### Three-class classification results of oral cancer cells

To determine which training model resulted in the best performance in the three-class classification, we first used the validation set from HOMO data set. As shown in Table 3, all models achieved above 98% mean accuracy, macro sensitivity, macro precision, macro F1 score, and macro AUC. Results of its test set (Table 4) were even better using EL; except for M3 model which achieved at least 99% in all metrics, other models showed a perfect score (100%). This also demonstrated the strength of the EL technique; otherwise, if each trained model (from five-fold CV) was evaluated separately on the test set (i.e., no EL applied), their averaged performances are similar to those for validation set, with accuracy ranging from 99.00% to 99.78%. Overall, these results suggested that EfficientNet-B3 demonstrated high proficiency in extracting the distinguished morphological characteristics of each class of cells and that the accuracy of the ensemble learning can be further improved by using all models.

**Table 4. The EL performances of CNN models on the test set of HOMO data.**

| Metrics (%) | Models | | | | |
|---|---|---|---|---|---|
| | M1 | M2 | M3 | M4 | M5 |
| *Accuracy* | **100.00** | **100.00** | 99.00 | **100.00** | **100.00** |
| *Macro Sensitivity* | **100.00** | **100.00** | 99.00 | **100.00** | **100.00** |
| *Macro Precision* | **100.00** | **100.00** | 99.00 | **100.00** | **100.00** |
| *Macro F1 score* | **100.00** | **100.00** | 99.00 | **100.00** | **100.00** |
| *Macro AUC* | **100.00** | **100.00** | 100.00 | **100.00** | **100.00** |

## Proportion approximation results of oral HETERO data

In addition to the HOMO dataset, we also used the HETERO dataset to examine the ability of EfficientNet-B3 to approximate the PARENTAL vs. RESISTENT cell ratio after model training with M1 to M5. We showed that the M1 model had a significantly higher mean error rate as estimated by RMSE than other models (validation set: 0.5926, test set: 0.5489), likely because M1 did not have HETERO training (Table 5). Meanwhile, while the M4 training data set resulted in the lowest error rate (0.2144), in the testing data set it was M5 that resulted in the lowest error rate compared (0.1592) to other models. This suggests that incorporating more HETERO data into model training and applying Random OS method indeed improved the performance of EfficientNet-B3. Further combination with DS method also helped generalize the model to unforeseen data. Notably, we found that the error rate of M3, which had Borderline-SMOTE preprocessing, is slightly higher than that of M2. Possibly, the synthetic data of M3 generated by linear interpolation may cause subtle distortion of the images to negatively affect the results. This was also reflected in its classification testing result in Table 3.

The superior performance of the M5 training data set was also revealed by analysis of probability distribution (in Tables 6 and 7). Indeed, when different ratios (including 1:2, 2:1, 1:3, and 3:1) of non-resistant and resistant cells were tested, M5 successfully depicted the proportional trends of cells in testing set (i.e., the estimated proportions of "RESISTANT" increased from data 1:2 to 1:3 and decreased from 2:1 to 3:1). Apart from the ratio approximation, we also examined the performance of models in identifying the dominant class in different data sets, as shown in Tables 8 and 9. Despite the diverse results in the training set, the M2, M3 and M5 models led to an accuracy above 90%, while M1 showed the lowest accuracy. Collectively, our results suggested that combining the HETERO data set with Random OS adequately improves model learning.

## Visual explanation with Grad-CAM results

To characterize the features that the models learned after training and determine whether the models could accomplish the classification task based on the shape of cells but not other

**Table 5. The performances (RMSE) of CNN models on HETERO data.**

| Data Sets | Models | | | | |
|---|---|---|---|---|---|
| | M1 | M2 | M3 | M4 | M5 |
| **Validation** | 0.5926 (0.0141) | 0.2182 (0.0167) | 0.2419 (0.0105) | **0.2144 (0.0099)** | 0.2182 (0.0123) |
| **Test** | 0.5489 | 0.1801 | 0.2119 | 0.1690 | **0.1592** |

For validation set, Mean(SE) was portrayed.

**Table 6. The five-fold performances (Mean(SE) of the approximated probabilities for PARENTAL: RESISTANT) of CNN models on validation set of HETERO data.**

| True Ratio (%) | Models | | | | |
|---|---|---|---|---|---|
| | **M1** | **M2** | **M3** | **M4** | **M5** |
| **1:2** | 85.82 (1.52): 11.53 (1.21) | 38.01 (3.19): 55.96 (4.03) | **37.67 (2.16): 56.63 (1.48)** | 38.04 (3.44): 56.23 (4.00) | 38.78 (5.75): 55.88 (5.72) |
| **2:1** | 61.88 (9.73): 34.23 (10.17) | 58.44 (2.97): 35.26 (3.18) | 57.32 (6.38): 36.14 (4.67) | **60.41 (2.15): 34.10 (2.84)** | 57.65 (7.66): 37.37 (6.86) |
| **1:3** | 66.98 (7.56): 29.65 (6.92) | 29.14 (1.92): 66.90 (1.80) | 33.91 (3.31): 61.86 (3.38) | 26.19 (2.50): 69.62 (2.72) | **23.88 (2.29): 72.71 (2.66)** |
| **3:1** | **87.66 (2.11): 10.33 (1.93)** | 53.37 (4.40): 42.91 (4.03) | 50.24 (2.74): 45.26 (2.57) | 54.02 (4.58): 42.21 (4.79) | 56.38 (4.32): 40.46 (4.75) |

The true ratios of HETERO data for classes– (PARENTAL, RESISTANT) are (33:66) for 1:2, (25:75) for 1:3, (20:80) for 1:4, and so on. Besides, the true ratio of class–CONTROL is kept 0.

**Table 7. The EL performances (approximated probabilities for PARENTAL: RESISTANT) of CNN models on the test set of HETERO data.**

| True Ratio (%) | Models | | | | |
|---|---|---|---|---|---|
| | **M1** | **M2** | **M3** | **M4** | **M5** |
| **1:2** | 80.89: 5.72 | 37.37: 56.55 | 37.95: 56.46 | 38.22: 55.70 | **35.80: 58.93** |
| **2:1** | 60.94: 35.16 | 58.06: 34.82 | 56.54: 37.08 | **59.49: 34.56** | 56.68: 37.83 |
| **1:3** | 62.40: 33.64 | 27.02: 68.77 | 32.85: 62.75 | 26.58: 69.29 | **24.17: 72.19** |
| **3:1** | 87.66: 10.45 | 56.52: 39.95 | 54.94: 41.01 | 58.86: 37.79 | **61.46: 35.58** |

**Table 8. The accuracies (Mean(SE)) of predicted dominant class of CNN models on validation set of HETERO data.**

| True Ratio (%) | Models | | | | |
|---|---|---|---|---|---|
| | **M1** | **M2** | **M3** | **M4** | **M5** |
| **1:2** | 3.12 (4.52) | 82.50 (13.18) | **93.12 (2.61)** | 83.12 (13.73) | 83.12 (17.62) |
| **2:1** | 68.75 (18.22) | 93.12 (4.64) | 83.12 (13.37) | **96.25 (5.13)** | 78.75 (15.37) |
| **1:3** | 29.09 (9.24) | 93.94 (3.71) | 86.67 (4.60) | 96.36 (3.95) | **96.36 (2.54)** |
| **3:1** | **95.48 (2.89)** | 65.81 (9.84) | 64.52 (12.90) | 69.68 (14.71) | 67.10 (12.37) |

The accuracies for 1:2 & 1:3 HETERO data are $Pr(RESISTANT > PARENTAL)$; on the other hand, they are $Pr(PARENTAL > RESISTANT)$ for 2:1 & 3:1 HETERO data.

**Table 9. The EL accuracies of predicted dominant class of CNN models on the test set of HETERO data.**

| True Ratio (%) | Models | | | | |
|---|---|---|---|---|---|
| | **M1** | **M2** | **M3** | **M4** | **M5** |
| **1:2** | 7.50 | **100.00** | **100.00** | 95.00 | **100.00** |
| **2:1** | 70.00 | 95.00 | 90.00 | **97.50** | 90.00 |
| **1:3** | 40.00 | **100.00** | 97.50 | **100.00** | **100.00** |
| **3:1** | **100.00** | 95.00 | 80.00 | 90.00 | 92.50 |

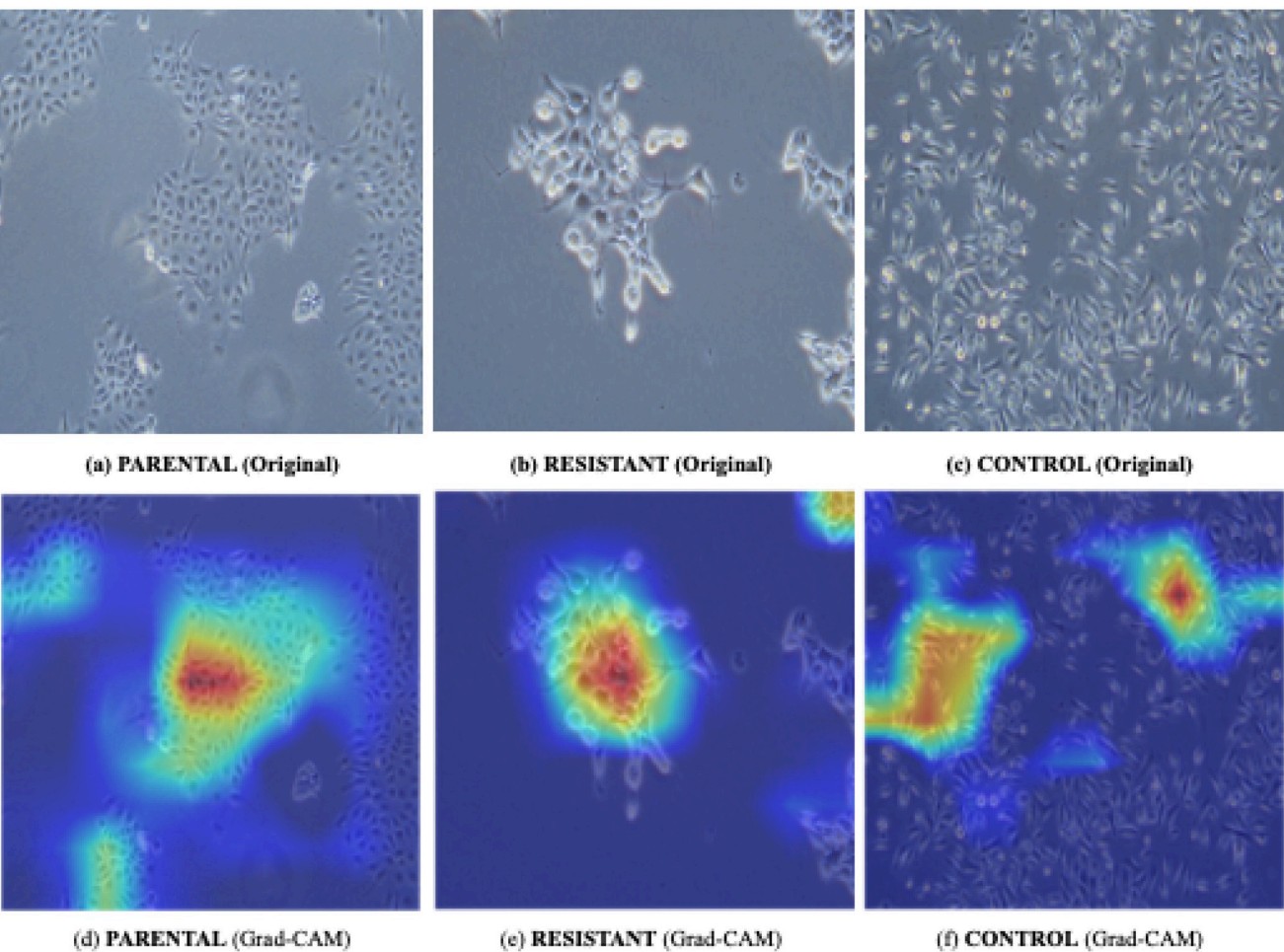

**Fig 7. An illustration of Grad-CAM results for three classes of HOMO test set.** (a) ∼ (c) are original images of the three classes, while (d) ∼ (f) are their corresponding Grad-CAM heat-maps. It showed that the model made decisions mainly by the important features (i.e., the contours of cell bodies) rather than trivial features (e.g., the gap between the cells).

features, the Grad-CAM results were used as a visual explanation to reveal how deep learning algorithms make decisions [39], aiming to elucidate the interpretability of our results in the context of chemoresistant oral cell morphology. While Grad-CAM primarily highlighted regions of importance for classification (such as cell contours, as evidenced in Fig 7), it is important to note that our study focuses on classification accuracy rather than direct prediction of the morphology of chemoresistant cells. The Grad-CAM heat maps effectively demonstrated how our models leverage significant features for classification tasks, which may only indirectly correlate with the underlying morphological characteristics critical for identifying chemoresistant cells. As shown in Fig 7, the representative Grad-CAM heat map of the M5 model in the three-class classification, the heat areas (yellow to red) indicated more weights the algorithm put for the final classification; while the model could not reason what morphology each type of cells should manifest, it did suggest that the model actually made decisions based mainly on important features, such as the contours of cell bodies themselves, rather than trivial features, such as the background color or the gap between cells. Overall, Gad-CAM demonstrated that although CNN learning trained with HETERO data was more of a regression task, it still performed well in the classification for HOMO data.

## Discussion

In this study, we used cell images from not only a single population but also a mixed population for CNN training as a regression rather than a classification task. We found that the CNN model can accomplish both the classification task and the probability approximation. Additionally, we used Grad-CAM to enhance the interpretability of the results by visually highlighting the regions in the images that contribute most significantly to the model's decisions, thereby indicating spatial features (i.e., contours of cell morphology) the model deems important for classification. These results support the notion that CNN-based image analysis remains useful for data featuring a phenotypic continuum, such as cell state transition during the acquisition of chemoresistance.

The key to CNN-based image analysis is the selection of the appropriate model and algorithm. Most CNN models were developed for ImageNet Large Scale Visual Recognition Challenge (ILSVRC), an annual computer vision competition [37] that used a large-scale imagery benchmark data set, called ImageNet, to evaluate the performance of competing algorithms in various visual recognition tasks, such as object detection and image classification. Early famous CNN models included AlexNet, VGG16, GoogLeNet, and ResNet, with a classification accuracy of 84.60%, 92.67%, 93.34%, and 94.29%, respectively [40–42]. In 2019, Google introduced EfficientNet, a revolutionary model that required fewer parameter inputs (i.e., lower model complexity) and yet achieved higher accuracy (97.1%) and efficiency (6.1x faster in computation) than previous CNN models [36]. Although more advanced models have been published later to further enhance accuracy, they often required many parameters and large data, and therefore, along model training time is usually needed [22, 43, 44].

In addition to a platform for evaluation of the performance of algorithms, ILSVRC also provides an opportunity for the networks developed for ImageNet to be applied to other types of image data. This process of knowledge transfer using the previous pre-trained model for the new task is called 'transfer learning'. Because transfer learning can directly utilize the pre-trained model architecture with useful initial feature extractors, it not only saves a huge amount of training time but also increases accuracy [35]. This feature is especially useful for medical data, which are often small in data size for CNN model training and usually require numerous parameters. For example, in our study we used only less than 2000 cell images for model training and obtained satisfactory results. Importantly, our study suggested that the EfficientNet-B3 framework can be applied to microscopic cell images with nearly indiscernible morphological variation between cells in varying states, which are very different from ImageNet, which comprises different types of images, achieving an accuracy of more than 98% in differentiating chemosensitive vs. chemoresistant cancer cells that are not visually distinguishable even by experts. This result is comparable to a previous study that used the VGG16 model to analyze drug-resistant colorectal cancers [45]. However, the EfficientNet-B3 model requires only 12 million parameters, whereas the VGG16 model requires a total of 138 million parameters, thus highlighting the strength of EfficientNet-B3 model selection. EfficientNet was also used to classify four subtypes of white blood cells (eosinophil, lymphocyte, monocyte, and neutrophil) for the early detection of diseases affecting specific subtypes (for example, lymphocytic leukocytosis, neutrophilic leukocytosis), with an accuracy of 90% [16, Chapter 4]. Despite the lower accuracy compared to our model, its approach prioritized speed by reducing the image dimensions to (width, height) = (60, 80). This highlights the trade-off in EfficientNet versions between accuracy and processing speed. Another study developed lightweight mobile CNN models (EfficientNet-B0, MobileNetV2 and NASNet Mobile) to classify B cell acute lymphoblastic leukemia [46]. The innovation was centered on a unique segmentation technique that

integrated a masking process to eliminate extraneous components in blood microscopy images, achieving a robust precision of 100%.

An issue commonly related to CNN modeling is the tendency to favor the major class prediction when a significant disparity of data between class distribution occurs or some classes have a very low proportion in data [47]. To deal with this potential bias, we applied data processing techniques. A popular method to balance different classes is to incorporate an OS method to generate more (synthetic) data for the minority class. Our results showed that the Random OS method, which is more intuitive and less complicated, outperformed the SMOTE method. Possibly, the make-up cell images that were generated by simple linear interpolation of two images using SMOTE or Mixup techniques were significantly distorted [48, 49], thereby negatively impacting the results. Alternatively, data can be accumulated from various resources to expand the size of the data. In this regard, the technique to leverage all sources of data with different scales, such as federated learning, should be used to avoid compromising the data set [50].

Although this study demonstrated that CNN-based image analysis can accomplish classification and approximation of a heterogeneous cell population, a limitation is the use of cancer cell lines rather than clinical tumor specimens. Indeed, while model training using cell lines with definitive chemoresistance is straightforward, this model cannot be extended to clinical application due to the lack of training with patient-derived tumor tissue. However, training a model to predict patient responses to chemotherapy is challenging due to the diversity of cancer cells within tumors, each potentially responding differently to chemotherapy, making it essentially impossible to label chemoresistant cancer cells among the population for model training. Nonetheless, since cancer cell lines are commonly used in the initial screening stage of drug development, serving as a reliable and consistent tool to assess the efficacy and toxicity of potential therapeutics, meticulous monitoring of these cells to detect altered drug responses is still valuable. Importantly, automated in vitro cell image analysis does not require cell processing, thereby enabling real-time assessment of cell state and functionality.

In conclusion, this study showed that CNN-based image analysis can accomplish classification and approximation of a heterogeneous cell population, which is a complex interplay of factors including the genetic diversity, tumor microenvironment, and cell adaptation. Meanwhile, this study also shows that an appropriate combination of CNN models and data processing techniques can compensate for the sample size deficiency. Further optimization with clinical samples should pave the way for future imaging diagnosis in clinical studies.

## Acknowledgments

The authors extend their gratitude to the editor and the four reviewers for their valuable feedback, which greatly enhanced the quality of the manuscript. We also wish to thank Prof. Shih-Hwa Chiou for supplying the data used in this research, Drs. Kao-Jung Chang and Kuo-Wei Chang for their valuable insights into data acquisition and analysis for our study. We are also grateful to the National Center for High-performance Computing for computer time and facilities.

## Author Contributions

**Conceptualization:** Kai-Feng Hung, Henry Horng-Shing Lu.

**Data curation:** Chung-Hsien Chou, Bou-Yue Peng, Yi-Chen Sun, Tzu-Wei Lin, Yueh Chien, Shih-Hwa Chiou, Kai-Feng Hung.

**Formal analysis:** Hsing-Chuan Hsieh.

**Funding acquisition:** Shih-Hwa Chiou, Henry Horng-Shing Lu.

**Investigation:** Hsing-Chuan Hsieh, Cho-Yi Chen, Kai-Feng Hung, Henry Horng-Shing Lu.

**Methodology:** Hsing-Chuan Hsieh, Henry Horng-Shing Lu.

**Software:** Hsing-Chuan Hsieh.

**Supervision:** Shih-Hwa Chiou, Henry Horng-Shing Lu.

**Validation:** Kai-Feng Hung.

**Writing – original draft:** Hsing-Chuan Hsieh.

**Writing – review & editing:** Cho-Yi Chen, Kai-Feng Hung, Henry Horng-Shing Lu.

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
