## [Decision Letter · Decision Letter 0]

16 May 2024

PONE-D-24-03873Automatic image classification for the morphological shape of oral cancer cells by deep learningPLOS ONE

Dear Dr. Lu,

Thank you for submitting your manuscript to PLOS ONE. After careful consideration, we feel that it has merit but does not fully meet PLOS ONE’s publication criteria as it currently stands. Therefore, we invite you to submit a revised version of the manuscript that addresses the points raised during the review process.

We look forward to receiving your revised manuscript.

Kind regards,

John Adeoye

Academic Editor

PLOS ONE

Journal Requirements:

 [National Science and Technology Council, Taiwan].  

[This work was supported in part by the National Science and Technology Council,

Taiwan, R.O.C., under Grant No. MOST-112-2634-F-A49-003,

MOST-112-2321-B-075-002, MOST-112-2321-B-A49-021, and 

MOST-110-2118-MA49-002-MY3 in part by the Higher Education Sprout Project of the 

National Yang Ming Chiao Tung University and Ministry of Education, Taiwan, R.O.C., 

and in part by Ministry of Education Yushan Scholar Program, Taiwan, R.O.C. We are 

grateful to the National Center for High-performance Computing for computer time and 

facilities.]

 [National Science and Technology Council, Taiwan]. 

6. We note that Figure(s) 1, 3, and 7 in your submission contain copyrighted images. All PLOS content is published under the Creative Commons Attribution License (CC BY 4.0), which means that the manuscript, images, and Supporting Information files will be freely available online, and any third party is permitted to access, download, copy, distribute, and use these materials in any way, even commercially, with proper attribution. For more information, see our copyright guidelines: http://journals.plos.org/plosone/s/licenses-and-copyright.

a. You may seek permission from the original copyright holder of Figure(s) 1, 3, and 7 to publish the content specifically under the CC BY 4.0 license. 

Additional Editor Comments:

Reviewers' comments:

Reviewer's Responses to Questions

**Comments to the Author**

1. Is the manuscript technically sound, and do the data support the conclusions?

Reviewer #1: Partly

Reviewer #2: Yes

Reviewer #3: Yes

2. Has the statistical analysis been performed appropriately and rigorously? 

Reviewer #1: Yes

Reviewer #2: Yes

Reviewer #3: Yes

3. Have the authors made all data underlying the findings in their manuscript fully available?

Reviewer #1: No

Reviewer #2: No

Reviewer #3: No

4. Is the manuscript presented in an intelligible fashion and written in standard English?

Reviewer #1: Yes

Reviewer #2: Yes

Reviewer #3: Yes

5. Review Comments to the Author

Reviewer #1: The authors applied a neural network to predict differences in morphological characteristics between two different cell lines (oral squamous cell carcinoma cells and a control kidney cell line -why not oral kerotinocytes as control?). Though their methodological design was thoughtful and creative, it has limited clinical significance since the experiments were performed in two cell lines and does not capture the histomorphologic spectrum of oral cancer cells.

Reviewer #2: This is a study about the use of deep learning in classification of the morphological shape of oral cancer cells. I have some suggestions to improve this manuscript.

Introduction

- Add the review of oral cancer cell morphology.

- Add the importance of using deep learning in classification of oral cancer cell morphology.

Discussion

- Discuss more about the previous studies in using machine learning/deep learning to classify cancer cell morphology.

- Compare the results of this study to related previous studies in classification of cancer cell morphology using machine learning/ deep learning.

Reviewer #3: The article utilizes convolutional neural networks (CNNs), particularly the EfficientNet-B3 model, to identify and categorize morphological changes in cancer cells, especially those induced by chemotherapy. The study successfully developed and validated a CNN model capable of accurately classifying normal cells, cancer cells, and chemotherapy-resistant cancer cells. Additionally, it can estimate the proportion of chemotherapy-resistant cells within a cell population. This research demonstrates the method's efficacy and precision in recognizing and categorizing cancer cells, particularly those resistant to chemotherapy, offering a novel approach for identifying and classifying subtle cellular morphological changes that are difficult to observe with traditional microscopy techniques. However, while the model exhibits high accuracy, its applicability and validation with actual clinical samples are insufficient. The study lacks an in-depth discussion on the model's applicability and effectiveness across different types of cancer cells. These limitations have also been demonstrated in the discussion section.

Further revisions are necessary for this paper:

1. Most of the literature cited in the introduction and discussion sections is from studies conducted over three years ago. It is advisable to include more recent references or update the citations.

2. Some of the cited literature comprises unreviewed preprints with a low level of evidence. It is recommended to replace these with references from more credible sources.

3. For the algorithm and modeling section, more explicit details about the hyperparameter tuning process would be beneficial. It's not clear how hyperparameters were optimized and what specific values were used.

4. The caption for Fig 1 requires a period at the end of the sentence.

“Fig 1. Three classes of cells (with labeling), named as HOMO data. Each class has equally 300 images. Notice that the morphological features of three classes of cells have only subtle differences that the expert might be able to distinguish.”

6. PLOS authors have the option to publish the peer review history of their article (what does this mean?). If published, this will include your full peer review and any attached files.

Reviewer #1: No

Reviewer #2: No

Reviewer #3: No

---

## [Author Response · Author response to Decision Letter 0]

25 Jul 2024

July 23, 2024

Dear Editor,

We are very grateful to have the opportunity to revise our manuscript entitled “Automatic image classification for the morphological shape of oral cancer cells in vitro by deep learning” (revised to “Deep learning-based automatic image classification of oral cancer cells acquiring chemoresistance in vitro”).

We greatly appreciate the reviewers’ comments and have revised the manuscript accordingly. Our responses are given in a point-by-point manner below. 

We hope our responses are satisfactory and the revised manuscript is appropriate for publication.

Thank you very much for your kind consideration of our manuscript.

Comments from Editor:

Comment #1:

Response #1:

We thank the Editor’s reminder and have ensured that our manuscript meets PLOS ONE’s requirements.

Comment #2:

Please note that PLOS ONE has specific guidelines on code sharing for submissions in which author-generated code underpins the findings in the manuscript. In these cases, all author-generated code must be made available without restrictions upon publication of the work. Please review our guidelines at https://journals.plos.org/plosone/s/materials-and-software-sharing#loc-sharing-code and ensure that your code is shared in a way that follows best practice and facilitates reproducibility and reuse.

Response #2:

We thank the Editor’s suggestion and have shared our code to facilitate reproducibility of our results.

Comment #3:

We note that the grant information you provided in the ‘Funding Information’ and ‘Financial Disclosure’ sections do not match. 

Response #3:

We thank the Editor’s suggestion and have corrected the grant numbers in the ‘Funding Information’ in cover letter.

Comment #4:

Thank you for stating the following financial disclosure: 

 [National Science and Technology Council, Taiwan]. 

Response #4:

We thank the Editor’s suggestion to clarity the role of the funder and have included the statement “The funders had no role in study design, data collection and analysis, decision to publish, or preparation of the manuscript” in cover letter.

Comment #5:

Thank you for stating the following in the Acknowledgments Section of your manuscript: 

[This work was supported in part by the National Science and Technology Council,

Taiwan, R.O.C., under Grant No. MOST-112-2634-F-A49-003,

MOST-112-2321-B-075-002, MOST-112-2321-B-A49-021, and 

MOST-110-2118-MA49-002-MY3 in part by the Higher Education Sprout Project of the 

National Yang Ming Chiao Tung University and Ministry of Education, Taiwan, R.O.C., 

and in part by Ministry of Education Yushan Scholar Program, Taiwan, R.O.C. We are 

grateful to the National Center for High-performance Computing for computer time and 

facilities.]

 [National Science and Technology Council, Taiwan]. 

Response #5:

We thank the Editor’s reminder to remove the funding information in the manuscript. We have updated the funding statement in the amended cover letter.

Comment #6:

We note that Figure(s) 1, 3, and 7 in your submission contain copyrighted images. All PLOS content is published under the Creative Commons Attribution License (CC BY 4.0), which means that the manuscript, images, and Supporting Information files will be freely available online, and any third party is permitted to access, download, copy, distribute, and use these materials in any way, even commercially, with proper attribution. For more information, see our copyright guidelines: http://journals.plos.org/plosone/s/licenses-and-copyright.

a. You may seek permission from the original copyright holder of Figure(s) 1, 3, and 7 to publish the content specifically under the CC BY 4.0 license. 

Response #6:

We thank the Editor for inquiring about the copyrights of the Figure(s) 1, 3, and 7. In fact, these figures are generated by our own team, and therefore, they are not subject to any copyright restrictions. They can be freely published under the Creative Commons Attribution License (CC BY 4.0), allowing unrestricted use, distribution, and reproduction with proper attribution.

Reviewer #1

Comment #1-1:

The authors applied a neural network to predict differences in morphological characteristics between two different cell lines (oral squamous cell carcinoma cells and a control kidney cell line - why not oral keratinocytes as control?). Though their methodological design was thoughtful and creative, it has limited clinical significance since the experiments were performed in two cell lines and does not capture the histomorphology spectrum of oral cancer cells.

Response #1-1:

We thank the reviewer’s comment. The primary aim of this study is to investigate whether CNN can effectively perceive changes in cell morphology, particularly when cells acquire features such as chemoresistance that may not be discernible through direct visual observation. While incorporating clinical specimens would undoubtedly enhance the value of the study, the heterogeneous nature of cancer cells poses challenges in clearly identifying chemotherapy-resistant cells within a tumor for model training. Despite this limitation, our study serves as a proof-of-concept demonstrating CNN's ability to discern cells with distinct features beyond what is perceptible to the naked eye. We believe that our findings are valuable in various applications; for example, CNN could serve as a tool for monitoring cell properties, thus facilitating quality control during cell culture. We acknowledge the reviewer's concerns regarding the use of a kidney cell line as the control. Therefore, in this revision, we have replaced the images of the kidney cell line with those of normal oral keratinocytes as the control for model training (Page 3, Lines 73, in red).

Reviewer #2:

This is a study about the use of deep learning in classification of the morphological shape of oral cancer cells. I have some suggestions to improve this manuscript.

Comment #2-1:

In introduction, add the review of oral cancer cell morphology.

Response #2-1:

We thank the reviewer’s suggestion and have included the review of oral cancer cell morphology (Page 2, Lines 48-53, in red).

Comment #2-2:

In introduction, add the importance of using deep learning in classification of oral cancer cell morphology.

Response #2-2:

We thank the reviewer’s suggestion to elaborate the importance of using deep learning in classification of cancer cell morphology, which lies in its ability to address challenges such as variability in cell morphology and error-prone manual feature extraction. Convolutional Neural Networks (CNN) excel at automatically identifying patterns and structures, making them ideal models for classifying cancer cell morphology, particularly those with subtle changes in cell shape. Moreover, deep learning-based automatic image classification also allows the determination of cell states without cell processing, thus extending its applicability to live cells. We incorporated these aspects, highlighting the importance of deep learning in the classification of cancer cell morphology in the Introduction (Page 3, Lines 40-58, in red).

Comment #2-3:

In Discussion, discuss more about the previous studies in using machine learning/deep learning to classify cancer cell morphology.

Response #2-3:

We thank the reviewer’s suggestion and have elaborated on previous studies that used deep learning for the classification of different types of cell morphology in the Discussion. (Page 14, Lines 424-434, in red).

Comment #2-4:

In Discussion, compare the results of this study to related previous studies in classification of cancer cell morphology using machine learning/ deep learning. 

Response #2-4: We thank the reviewer’s suggestion and have incorporated comparisons with other studies, including those that prioritized the speed of the process by reducing the dimensions of the image, integrated masking processes to develop lightweight models, and used the VGG16 model to improve precision. (Page 13, Lines 424-434, in red). 

Reviewer #3:

Comment #3-1:

The article utilizes convolutional neural networks (CNNs), particularly the EfficientNet-B3 model, to identify and categorize morphological changes in cancer cells, especially those induced by chemotherapy. The study successfully developed and validated a CNN model capable of accurately classifying normal cells, cancer cells, and chemotherapy-resistant cancer cells. Additionally, it can estimate the proportion of chemotherapy-resistant cells within a cell population. This research demonstrates the method's efficacy and precision in recognizing and categorizing cancer cells, particularly those resistant to chemotherapy, offering a novel approach for identifying and classifying subtle cellular morphological changes that are difficult to observe with traditional microscopy techniques. However, while the model exhibits high accuracy, its applicability and validation with actual clinical samples are insufficient. The study lacks an in-depth discussion on the model's applicability and effectiveness across different types of cancer cells. These limitations have also been demonstrated in the discussion section.

Response #3-1:

We thank the reviewer for highlighting the limitation of lacking clinical significance and recommending a discussion on the applicability of the model. We realize that only the model that has been trained with clinical tumor specimens, but not cancer cell lines, can be extended to clinical application. However, training a model to predict patient responses to chemotherapy is challenging because it is essentially impossible to label chemoresistant cancer cells within tumors for model training. Nonetheless, a potential applicability of the model is related to the screening stage of drug development because cancer cell lines are commonly used as a reliable and consistent tool for assessing the efficacy and toxicity of the drugs. Additionally, we also mentioned the value of automated in vitro cell image analysis for real-time assessment of cell state and functionality. We have included these notions in the Discussion (Page 14, Lines 447-461, in red).

Comment #3-2:

Most of the literature cited in the introduction and discussion sections is from studies conducted over three years ago. It is advisable to include more recent references or update the citations.

Response #3-2:

We thank the reviewer’s suggestion and have incorporated more recent references.

Comment #3-3:

Some of the cited literature comprises unreviewed preprints with a low level of evidence. It is recommended to replace these with references from more credible sources.

Response #3-3:

We thank the reviewer’s suggestion and have updated the references.

Comment #3-4:

For the algorithm and modeling section, more explicit details about the hyperparameter tuning process would be beneficial. It's not clear how hyperparameters were optimized and what specific values were used.

Response #3-4:

We thank the reviewer’s suggestion and have summarized the hyperparameter tuning process as well as the hyperparameter setting in a new section “Hyperparameters Setting”. Two tables are also included to increase its clarity. (Page 7, lines 247-280, Table 1 and Table 2, in red).

Comment #3-5:

The caption for Fig 1 requires a period at the end of the sentence.

“Fig 1. Three classes of cells (with labeling), named as HOMO data. Each class has equally 300 images. Notice that the morphological features of three classes of cells have only subtle differences that the expert might be able to distinguish.”

Response #3-5:

We thank the reviewer’s suggestion and have corrected the error in according to the instruction. (Page 4, Fig 1.).

Reviewer 4

The research investigates the use of CNN to distinguish between different cell types and hopes to identify chemo-resistant cancer cells based on their morphological features. The manuscript has potential but needs further work to be published. Please revise, clarify, or address the following:

Comment #4-1:

The current title may be construed as misleading since it implies that the image analysis is completed on oral cancer tissue and inflates the clinical significance of the paper. Usage of cell lines for image analysis is also not mentioned at all in the abstract.

Response #4-1:

We thank the reviewer for pointing out that the current title misleads by implying the use of oral cancer tissue for the experiments. We have revised the title to “Deep learning-based automatic image classification of oral cancer cells acquiring chemoresistance in vitro”. We also revised the Abstract to explicitly mention the use of oral cancer cell lines in this study (Page 1, in red).

Comment #4-2:

HEK293T cells (“CONTROL”) – why choose kidney cells for control instead of oral keratinocytes?

Response #

---

## [Decision Letter · Decision Letter 1]

29 Aug 2024

Deep learning-based automatic image classification of oral cancer cells acquiring chemoresistance in vitro

PONE-D-24-03873R1

Dear Dr. Lu,

We’re pleased to inform you that your manuscript has been judged scientifically suitable for publication and will be formally accepted for publication once it meets all outstanding technical requirements.

Kind regards,

John Adeoye

Academic Editor

PLOS ONE

Additional Editor Comments (optional):

Reviewers' comments:

Reviewer's Responses to Questions

**Comments to the Author**

1. If the authors have adequately addressed your comments raised in a previous round of review and you feel that this manuscript is now acceptable for publication, you may indicate that here to bypass the “Comments to the Author” section, enter your conflict of interest statement in the “Confidential to Editor” section, and submit your "Accept" recommendation.

Reviewer #2: All comments have been addressed

Reviewer #3: All comments have been addressed

2. Is the manuscript technically sound, and do the data support the conclusions?

Reviewer #2: Yes

Reviewer #3: Yes

3. Has the statistical analysis been performed appropriately and rigorously? 

Reviewer #2: Yes

Reviewer #3: Yes

4. Have the authors made all data underlying the findings in their manuscript fully available?

Reviewer #2: Yes

Reviewer #3: Yes

5. Is the manuscript presented in an intelligible fashion and written in standard English?

Reviewer #2: Yes

Reviewer #3: Yes

6. Review Comments to the Author

Reviewer #2: The authors have addressed reviewers' comments. The manuscript is improved and ready for the publication.

Reviewer #3: The manuscript has been revised according to the reviewers' comments, and I agree to the publication of this paper.

7. PLOS authors have the option to publish the peer review history of their article (what does this mean?). If published, this will include your full peer review and any attached files.

Reviewer #2: No

Reviewer #3: No

---

## [Editor Report · Acceptance letter]

25 Sep 2024

PONE-D-24-03873R1 

PLOS ONE

Dear Dr. Lu, 

I'm pleased to inform you that your manuscript has been deemed suitable for publication in PLOS ONE. Congratulations! Your manuscript is now being handed over to our production team.

Kind regards, 

on behalf of

Dr. John Adeoye 

Academic Editor

PLOS ONE